# Impact of supply chain disruptions and drug shortages on drug utilization: A scoping review protocol

Araniy Santhireswaran[1]*, Martin Ho[1], Kaitlin Fuller[2], Etienne Gaudette[3], Lisa Burry[1,4], Mina Tadrous[1]

1 Leslie Dan Faculty of Pharmacy, University of Toronto, Toronto, Canada, 2 Angus L. Macdonald Library, St. Francis Xavier University, Antigonish, Nova Scotia, Canada, 3 Institute of Health Policy, Management and Evaluation, University of Toronto, Toronto, Ontario, Canada, 4 Lunenfeld-Tanenbaum Research Institute, Mount Sinai Hospital, Sinai Health, Toronto, Ontario, Canada

* araniy.santhireswaran@mail.utoronto.ca

## Abstract

### Objective

This proposed scoping review aims to examine studies assessing the impact of drug shortages on population-level drug utilization trends. The objectives of this review are to a) assess which drugs have been studied and describe associated drug characteristics, b) determine jurisdictions and healthcare settings that have conducted these studies, and c) describe how changes in drug use and the extent of shortage impacts are reported in literature.

### Introduction

Drug shortages continue to impair drug access and delivery of quality care across the world. However, the impact of drug supply disruptions on availability and drug use are understudied in current literature. This proposed scoping review will identify this gap and inform future research initiatives aimed at determining the real-world impacts of drug shortages.

### Inclusion criteria

Published and unpublished observational studies reporting on the effects of drug supply chain disruptions (shortages, discontinuations, and safety-based withdrawals) on consequent utilization trends faced by pharmaceutical products (i.e. prescription drugs, over-the-counter drugs, vaccines, therapy products, pharmaceutical solutions). Literature reviews, meta-analyses, randomized control trials, case series, case reports, and opinion pieces will be excluded.

### Methods

The search strategy will combine two key search concepts: drug shortages and drug utilization. The search will be conducted in MEDLINE and EMBASE. This will be followed by an extensive grey literature search in grey literature databases, targeted websites and Google.

**Data Availability Statement:** No datasets were generated or analysed during the current study. All relevant data from this study will be made available in the final publication upon study completion.

**Funding:** The author(s) received no specific funding for this work.

**Competing interests:** The authors have declared that no competing interests exist.

Furthermore, reference lists of included articles will be searched. Articles will be independently screened, selected and extracted by two reviewers. Data will be descriptively analyzed and presented in tables.

## Trial registration

**Review registration number**: Open Science Framework, https://osf.io/2p6e5.

## Introduction

Drug shortages are a growing concern in health systems across the world. Over 13,000 drugs were at risk for shortage over the last five years, and of these almost half faced at least one shortage [1]. The World Health Organization (WHO) defines a drug shortage as a situation where supplies of essential medicines, health products, and vaccines are insufficient to meet public health and patient needs [2]. Drug shortages can manifest as temporary delays in drug availability or even permanent drug discontinuations. They can be caused by many reasons, including manufacturing problems, quality assurance issues, sole-source contracts, demand increases, and difficulty obtaining raw materials [3–5]. To mitigate the impact of shortages on the healthcare system, various strategies have been implemented, including mandatory reporting, changes to drug policies, expediting drug approval, and facilitating external importation [1,5–7]. Even with these efforts, however, more than 46,000 shortages were reported across 14 countries between 2017 and 2019, representing a 60% increase in shortages during that time period [8]. This illustrates the importance of this global problem and the need for further action to conserve access to essential medicines [4,5].

Drug shortages have a wide swath of potential implications for the healthcare system and care delivery [1]. The lack of medication availability leads to treatment alteration, switching, or discontinuation, which can all negatively influence patient outcomes. These include increased adverse effects such as toxicity of the alternative treatment, increased medication errors, patient non-adherence, inferior treatment, hospitalization, and even mortality [1,4,5,9–14]. For example, the recent valsartan recall led to a 15.7% global decrease in drug use [15], signifying challenges with continuing therapy, and in turn, negatively affecting clinical outcomes and further burdening the healthcare system. This stresses the urgent need for action by the government and stakeholders to undertake research initiatives aimed at better understanding the impact of drug shortages as a means of mitigating and ultimately preventing them.

Leveraging real-world data to examine population-level drug use trends can provide insight into the mechanisms behind the drug supply chain and shortages. These insights can further inform drug shortage frameworks and policy-making decisions. However, there is limited research on the impact of drug shortages on population-level drug use and access. A recent scoping review examining all literature related to drug shortages determined that out of 430 articles only 50 papers were retrospective or observational in nature [16]. Therefore, there may be an important knowledge gap in real-world drug utilization patterns during shortages.

To approach this gap in literature, we propose a scoping review of observational studies that examine drug utilization trends due to shortages. This proposed scoping review aims to 1) assess which drugs have been studied during times of shortage and describe associated drug characteristics, 2) determine the jurisdictions and healthcare settings that have conducted these studies and 3) describe how changes in drug use and the extent of impact are reported. A scoping review methodology is appropriate to address these aims by examining a wide range

of literature to understand how studies assess and report on drug use during shortages [17]. Moreover, the range of a scoping review helps identify the breadth of evidence available on the topic, alongside important gaps [18].

A preliminary search was conducted on MEDLINE, JBI Evidence Synthesis, Open Science Framework, Prospero, and the Cochrane Database of Systematic Reviews, and no current or underway scoping or systematic reviews on this topic were identified as of August 3$^{rd}$, 2023. Findings from this proposed scoping review will further the understanding of the impact of drug shortages on real-world drug use and access.

## Review questions

How is the impact of pharmaceutical product shortages on utilization trends being studied in literature?

a.  What drugs have been studied and what are associated drug characteristics?

b.  What jurisdictions and healthcare settings have studies been conducted in?

c.  How are changes in drug use being reported and to what extent is the impact described?

## Eligibility criteria

**Participants.**   This review will include pharmaceutical products including prescription drugs, over-the-counter drugs, vaccines, therapy products, pharmaceutical solutions, and sterile solutions, that have encountered a disruption such as shortages, discontinuations, recalls, and safety-based drug withdrawals.

**Concept.**   This proposed scoping review will consider drug supply chain disruptions including shortages, discontinuations, recalls and safety-based withdrawals, and their impact on drug access and use.

**Context.**   This review will examine studies using population-level drug utilization data from all healthcare settings and regions.

**Types of sources.**   For this scoping review, published and unpublished observational studies such as cohort, case-control and self-controlled designs, will be included. Our review question focuses on the impacts of drug shortages on drug use in large populations. Therefore, case series and case reports will be excluded since they only recount a few occurrences of the outcome instead of analyzing large populations. Moreover, randomized control trials (RCTs) will be excluded given that randomization is infeasible for our review question, and we aim to better understand the ral-world impact of shortages. Literature reviews (i.e. systematic, scoping, narrative, etc.), meta-analyses, study protocols, opinion pieces (i.e. editorials, commentaries, letters, etc.), conference abstracts, and studies with unavailable full-text articles will be excluded due to the lack of primary research analysis. A grey literature search will be conducted to include unpublished observational studies and reports from government organizations and healthcare organizations.

## Methods

This review will follow the JBI approach to scoping reviews which was built upon the six steps first articulated by Arksey & O'Malley (2005) and then further enhanced by Levac and colleagues (2012) [19–21]. The final paper will follow the PRISMA extension for Scoping Reviews (PRISMA ScR) to follow proper reporting standards [22].

## Information sources

A structured search will be conducted in MEDLINE and EMBASE. A grey literature search will be conducted using grey literature databases, a targeted website search, and the Google search engine. We understand that searching 3 databases is the standard for a scoping review. However, given that our search strategy is very specific we decided to conduct an extensive grey literature search instead of including a third broad interdisciplinary database. We conducted preliminary explorations of our search in MEDLINE, EMBASE, Web of Science, Scopus and Pharmaceutical abstracts and decided that both MEDLINE and EMBASE were sufficient for our review. Given our niche topic, interdisciplinary databases such as Web of Science, Scopus and Pharmaceutical Abstracts did not bring in many relevant articles. We acknowledge that including only 2 databases is a potential limitation of our scoping review. Additionally, a supplementary hand-search of reference lists of included articles and any relevant reviews identified in screening will be conducted to identify any additional relevant articles that may have been missed by the search strategy.

## Search strategy

In developing the search strategy, a preliminary limited search of MEDLINE was conducted to identify relevant articles on the topic. The preliminary search advised the selection of the key search concepts used in our search strategy which includes drug shortages and drug utilization. The titles and abstracts of relevant articles were scanned to select subject headings and text words used to develop the search strategy for MEDLINE (see S1 Appendix). The subject headings, textword and keyword queries as well as other database specific syntax will be adapted for adapted for EMBASE.

A grey literature search will be conducted using grey literature databases, a targeted website search and the google search engine. Grey literature databases such as World Health Organization (WHO) Data Collections, Health Services Research Projects in Progress, New York Academy of Medicine's Grey Literature Report, and Trip Pro will be searched, and relevant items will be selected within the database search output. The targeted website search will be conducted using Advanced Search on Google to filter for relevant uploaded reports and files. A list of websites is provided (see S2 Appendix). Many government and healthcare organizations lead drug shortage related research aimed at guiding policy decisions; therefore, government and healthcare websites were primarily included in the targeted website search. The Google search engine will be searched with five different queries reflecting the key search concepts, and the first five pages will be scanned by an independent reviewer to located potentially relevant items.

## Study records

**Data management.** After running the search in both MEDLINE and EMBASE, all identified citations will be imported to EndNote 20.5 Desktop (Clarivate Analytics, PA, USA) for screening. The Bramer method of de-duplication will be used to remove all duplicate citations [23]. Covidence Systematic Review Software (Veritas Health Innovation, Melbourne, Australia) will be used to facilitate the study selection process. Microsoft Excel will be used for the data collection process.

**Selection process.** A pilot screening will be conducted together by two independent reviewers to ensure that both reviewers understand the inclusion criteria and to ensure consistency. First, a test screen using a subset of articles will be conducted by two independent reviewers to ensure accuracy. The titles and abstracts of articles that do not fit the inclusion criteria will be removed, and any disagreements or adjustments to the eligibility criteria that arise

will be discussed by the reviewers. Microsoft Excel will be used to calculate the interrater agreement between the two independent reviewers. If 80% agreement is reached between the two reviewers, the remaining articles will be screened independently by the two reviewers. If the percent agreement is less than 80%, a new subset of articles will be screened until adequate agreement is reached [24]. The full-text for articles that potentially meet the criteria will be downloaded and imported into Covidence Systematic Review Software) for full-text screening.

The full texts of the articles that passed the first screening will be examined in detail against the inclusion criteria by two independent reviewers. If an article does not meet the inclusion criteria, the reason for exclusion will be recorded. Any disagreements between the reviewers will be discussed and resolved with a third reviewer. The results of the search and study selection will be presented in the final scoping review in a Preferred Reporting Items for Systematic Reviews and Meta-analyses extension for scoping review (PRISMA-ScR) flow diagram [22].

**Data collection process.** A data extraction form will be created using Microsoft Excel to record necessary information from all included articles. Prior to extraction, both reviewers will pilot the data extraction form on a subset of articles to ensure accurate and consistent extraction of data. After a consensus is reached, extraction of the remaining articles will be performed independently by both reviewers on the same included articles. To ensure consistency, a third reviewer will spot check 5% of the included articles.

**Data items.** The extracted data variables will include general article characteristics (i.e., year, authors, journal, etc.), study design, data sources, drug characteristics (i.e., formulation, drug class, dosage), shortage characteristics (i.e., duration, reason, status), outcomes (i.e., changes in drug use), key findings, conclusions, and funding sources. A draft extraction tool is provided (see S3 Appendix). Throughout the process of extraction, the data extraction form will be modified as necessary, and modification will be reported in the final scoping review. Any disagreements between the two reviewers will be resolved through discussion and/or with an additional reviewer. In any case of missing data, the authors of the paper will be contacted and requested to provide additional information.

## Data analysis and presentation

A descriptive analysis and quantitative analysis of the included articles will be conducted using Microsoft Excel, alongside a qualitative analysis of appropriate variables. Data for the first review question will detail frequency counts of drug characteristics of those discussed in studies, such as drug class, formulation, shortage status (i.e., discontinuation, recall, shortage, etc.), market structure, etc. Data answering the second review question will reflect frequency counts of the healthcare setting (i.e., hospital, community) and region the study was conducted in. For the third review question, data will outline frequency counts of the ways quantitative changes in drug use are reported and report the range of the changes in drug use. The results of search and study inclusion process will be displayed in a PRISMA flow diagram [22]. Relevant data from the extraction will be presented in appropriate tables and figures to showcase existing evidence and emphasize knowledge gaps in literature. All data will be made available in the supplemental tables of the final published manuscript. The results from this scoping review will guide future observational studies to further understand the impact of drug supply disruptions on population-level drug utilization and access.

## Significance

Drug shortages continue to impair care delivery, patient clinical outcomes and the healthcare system. Understanding the extent, frequency and severity of drug shortages can help inform important policy decisions aimed at mitigating shortages. Currently there is limited research

on the impact of drug shortages on patient drug use and access, and this proposed scoping review will summarize commonly studied shortages and their associated characteristics, settings and regions these studies have been conducted and most importantly the extent of the impact on use and access. It is crucial to compare the impacts of different shortages since not all shortages manifest similarly and have equal impacts. The findings from this proposed scoping review will inform future drug shortage research aimed at guiding drug shortage frameworks and policy-making decisions.

## Supporting information

**S1 Appendix. Search strategy.**
(DOCX)

**S2 Appendix. Targeted website search.**
(DOCX)

**S3 Appendix. Data extraction form.**
(DOCX)

**S4 Appendix. PRISMA scoping review checklist.**
(DOCX)

## Author Contributions

**Conceptualization:** Araniy Santhireswaran, Etienne Gaudette, Lisa Burry, Mina Tadrous.

**Investigation:** Araniy Santhireswaran.

**Methodology:** Araniy Santhireswaran, Kaitlin Fuller.

**Project administration:** Araniy Santhireswaran, Martin Ho.

**Supervision:** Kaitlin Fuller, Mina Tadrous.

**Validation:** Araniy Santhireswaran, Mina Tadrous.

**Visualization:** Araniy Santhireswaran, Kaitlin Fuller, Mina Tadrous.

**Writing – original draft:** Araniy Santhireswaran.

**Writing – review & editing:** Araniy Santhireswaran, Martin Ho, Kaitlin Fuller, Etienne Gaudette, Lisa Burry, Mina Tadrous.

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
