## [Decision Letter · Decision Letter 0]

25 Jun 2024

PONE-D-23-29159Impact of supply chain disruptions and drug shortages on drug utilization: a scoping review protocolPLOS ONE

Dear Dr. Santhireswaran,

Thank you for submitting your manuscript to PLOS ONE. After careful consideration, we feel that it has merit but does not fully meet PLOS ONE’s publication criteria as it currently stands. Therefore, we invite you to submit a revised version of the manuscript that addresses the points raised during the review process.

We look forward to receiving your revised manuscript.

Kind regards,

Udoka Okpalauwaekwe, MD, MPH, PhD

Academic Editor

PLOS ONE

Journal Requirements:

Reviewers' comments:

Reviewer's Responses to Questions

**Comments to the Author**

1. Does the manuscript provide a valid rationale for the proposed study, with clearly identified and justified research questions?

Reviewer #1: Yes

Reviewer #2: Yes

2. Is the protocol technically sound and planned in a manner that will lead to a meaningful outcome and allow testing the stated hypotheses?

Reviewer #1: Yes

Reviewer #2: Partly

3. Is the methodology feasible and described in sufficient detail to allow the work to be replicable?

Reviewer #1: Yes

Reviewer #2: No

4. Have the authors described where all data underlying the findings will be made available when the study is complete?

Reviewer #1: No

Reviewer #2: No

5. Is the manuscript presented in an intelligible fashion and written in standard English?

Reviewer #1: Yes

Reviewer #2: Yes

6. Review Comments to the Author

You may also provide optional suggestions and comments to authors that they might find helpful in planning their study.

Reviewer #1: I read with interest the protocol for a scoping a review written by Santhireswaran et al. The protocol describes methods for a scoping review to look at observational studies effects of drug supply chain disruptions. Overall, the manuscript is well written and describes the process adequately. I have a few recommendations and concerns however:

1. The authors should describe in more detail the logic of only including observational studies and excluding RCTS, etc. I would anticipate most of the studies will be observational in nature but it would be beneficial to the reader to understand he logic of excluding other types of studies as they have mentioned.

2. The authors have not clearly stated how the data will be made available in the future

3. Citations: There are certain times when the citations are placed before the period of a sentence and other times where it is placed after. Please revise this to keep it uniform and in line with PLOS formatting.

4. My preference would be that the protocol ends with the brief discussion and impact session to once again revision the potential impact of this work (i.e how it will change practice and/or policy). I believe this would strengthen the the protocol. The authors already provide some of this information in the introduction section but having it at the end of the paper may be of benefit too.

Reviewer #2: 1) While literature reviews are listed in the exclusion criteria, will literature reviews be checked to see if they include any eligible observational studies that may have been missed by the search strategy?

2) Only searching MEDLINE and EMBASE may not be adequate for a comprehensive scoping review. The search strategy would be improved by searching at least one broad interdisciplinary database such as Scopus and at least one pharmaceutical literature database such as the International Pharmaceutical Abstracts database.

3) The grey literature search strategy requires further development and explanation. What is the rationale for including each grey literature source? Why are preprints and conference proceedings not being searched as part of the grey literature strategy?

7. PLOS authors have the option to publish the peer review history of their article (what does this mean?). If published, this will include your full peer review and any attached files.

Reviewer #1: No

Reviewer #2: **Yes: **Gregory Laynor

---

## [Author Response · Author response to Decision Letter 0]

10 Jul 2024

July 10th, 2024

Dr. Udoka Okpalauwaekwe

Academic Editor

PLOS One

Re: PLOS ONE - Decision on Manuscript PONE-D-23-29159

Dear Dr. Okpalauwaekwe, 

We are delighted to learn of your journal’s interest in publishing our work and are grateful for the further suggestions provided. We have made the suggested changes to the manuscript based on the thoughtful suggestions from the reviewer’s comments. A point-by-point response to the comments is provided below. 

Sincerely, 

Araniy Santhireswaran

Reviewer Comments to the Author

Reviewer #1

1. The authors should describe in more detail the logic of only including observational studies and excluding RCTS, etc. I would anticipate most of the studies will be observational in nature but it would be beneficial to the reader to understand the logic of excluding other types of studies as they have mentioned.

Thank you for this suggestion, we agree that explaining our reasoning will be helpful for readers, so we have added language about that in the eligibility criteria section and more clearly communicated our reasoning.

Change: For this scoping review, published and unpublished observational studies such as cohort, case-control and self-controlled designs, will be included. Our review question focuses on the impacts of drug shortages on drug use in large populations. Therefore, case series and case reports will be excluded since they only recount a few occurrences of the outcome instead of analyzing large populations. Moreover, randomized control trials (RCTs) will be excluded given that randomization is infeasible for our review question, and we aim to better understand the real-world impact of shortages. Literature reviews (i.e. systematic, scoping, narrative, etc.), meta-analyses, study protocols, opinion pieces (i.e. editorials, commentaries, letters, etc.), conference abstracts, and studies with unavailable full-text articles will be excluded due to the lack of primary research analysis.

2. The authors have not clearly stated how the data will be made available in the future

Thank you for pointing this out. We added a statement about data availability and highlighted this in our data disclosure.

Change: Relevant data from the extraction will be presented in appropriate tables and figures to showcase existing evidence and emphasize knowledge gaps in literature. All data will be made available in the supplemental tables of the final published manuscript or in an OSF project space with the link shared in the manuscript.

3. Citations: There are certain times when the citations are placed before the period of a sentence and other times where it is placed after. Please revise this to keep it uniform and in line with PLOS formatting.

Thank you for bringing this to our attention, we have ensured the superscript formatting is consistent throughout the manuscript.

4. My preference would be that the protocol ends with the brief discussion and impact session to once again revision the potential impact of this work (i.e how it will change practice and/or policy). I believe this would strengthen the protocol. The authors already provide some of this information in the introduction section but having it at the end of the paper may be of benefit too.

This is a great suggestion, thank you. We have now incorporated a significance section at the end of the manuscript.

Added: 

Significance 

Drug shortages continue to impair care delivery, patient clinical outcomes and the healthcare system. Understanding the extent, frequency and severity of drug shortages can help inform important policy decisions aimed at mitigating shortages. Currently there is limited research on the impact of drug shortages on patient drug use and access, and this proposed scoping review will summarize commonly studied shortages and their associated characteristics, settings and regions these studies have been conducted and most importantly the extent of the impact on use and access. It is crucial to compare the impacts of different shortages since not all shortages manifest similarly and have equal impacts. The findings from this proposed scoping review will inform future drug shortage research aimed at guiding drug shortage frameworks and policy-making decisions. 

Reviewer #2

1. While literature reviews are listed in the exclusion criteria, will literature reviews be checked to see if they include any eligible observational studies that may have been missed by the search strategy?

Thank you for this suggestion. We will be checking the references list for some selected literature reviews in the field as a part of the reference hand search. We have updated this in the methods as well.

Change: A structured search will be conducted in MEDLINE and EMBASE. A grey literature search will be conducted using grey literature databases, a targeted website search, and the Google search engine. A supplementary hand-search of reference lists of included articles and any relevant reviews identified in screening will be conducted to identify any additional relevant articles that may have been missed by the search strategy. 

2. Only searching MEDLINE and EMBASE may not be adequate for a comprehensive scoping review. The search strategy would be improved by searching at least one broad interdisciplinary database such as Scopus and at least one pharmaceutical literature database such as the International Pharmaceutical Abstracts database.

Thank you for this comment. We acknowledge that three databases are the standard for a scoping review. However, given that our question is very specific and will rely on a data collection strategy that goes beyond published literature, we decided that adding a broad interdisciplinary database, like Scopus or Web of Science, would make our published literature searches too sensitive for this question and we wanted to focus more of our data collection efforts on the grey literature searches. We felt that the two largest biomedical databases MEDLINE and EMBASE, which has the pharmaceutical focus, combined with TRIPPro which was included as part of our grey literature search, but does search for published literature combined with the supplemental reference tracking searches will be sufficient for gathering the published literature for this scoping review. We consulted our librarian who is also a co-author and tested searches in Web of Science and Pharmaceutical Abstracts to arrive at this conclusion. We felt that this search strategy, that has more of an emphasis on grey literature, compared with the guidance for a standard systematic review search, allows us to more fulsomely search for and answer our question. However, we realize that the lack of a third interdisciplinary database is a limitation and we have made changes to make this clear.

Change: A structured search will be conducted in MEDLINE and EMBASE. A grey literature search will be conducted using grey literature databases, a targeted website search, and the Google search engine. We understand that searching 3 databases is the standard for a scoping review and including a broad interdisciplinary database will improve our search. However, given that our search strategy is very specific we decided to conduct an extensive grey literature search instead of including a third broad interdisciplinary database. We acknowledge that this is a potential limitation of our scoping review.

3. The grey literature search strategy requires further development and explanation. What is the rationale for including each grey literature source? Why are preprints and conference proceedings not being searched as part of the grey literature strategy? 

Thank you for these suggestions. For the purposes of this grey literature search, we want to focus on finding the research disseminated through alternative means of publication, for example, the reports that many government and healthcare organizations may produce but will not publish in a traditional article and the search strategy was designed with this in mind. We have elaborated on the rationale for the grey literature search by explaining why we chose these sources. As for your suggestion about preprints, the papers we aimed at including are not deposited in pre-print servers so we felt that searching these would not provide many relevant studies. Also, conference abstracts to be included as a standalone are not substantial enough as they lack the details of a full paper and can often present preliminary data that could change by the time the research is completed. Therefore, our grey literature databases focus on studies conducted by government and healthcare organizations primarily, and we have made changed to further explain this in the manuscript.

Change: The targeted website search will be conducted using Advanced Search on Google to filter for relevant uploaded reports and files. A list of websites is provided (see Appendix 2). Many government and healthcare organizations lead drug shortage related research aimed at guiding policy decisions; therefore, government and healthcare websites were primarily included in the targeted website search.

---

## [Decision Letter · Decision Letter 1]

23 Jul 2024

PONE-D-23-29159R1Impact of supply chain disruptions and drug shortages on drug utilization: a scoping review protocolPLOS ONE

Dear Dr. Santhireswaran,

Thank you for submitting your manuscript to PLOS ONE. After careful consideration, we feel that it has merit but does not fully meet PLOS ONE’s publication criteria as it currently stands. Therefore, we invite you to submit a revised version of the manuscript that addresses the points raised during the review process. Kindly pay close attention to the comments from Reviewer 2 and provide rationale and further clarification in your protocol.

We look forward to receiving your revised manuscript.

Kind regards,

Udoka Okpalauwaekwe, MD, MPH, PhD

Academic Editor

PLOS ONE

Journal Requirements:

Reviewers' comments:

Reviewer's Responses to Questions

**Comments to the Author**

1. Does the manuscript provide a valid rationale for the proposed study, with clearly identified and justified research questions?

Reviewer #1: Yes

Reviewer #2: Yes

2. Is the protocol technically sound and planned in a manner that will lead to a meaningful outcome and allow testing the stated hypotheses?

Reviewer #1: Yes

Reviewer #2: Partly

3. Is the methodology feasible and described in sufficient detail to allow the work to be replicable?

Reviewer #1: Yes

Reviewer #2: Yes

4. Have the authors described where all data underlying the findings will be made available when the study is complete?

Reviewer #1: Yes

Reviewer #2: Yes

5. Is the manuscript presented in an intelligible fashion and written in standard English?

Reviewer #1: Yes

Reviewer #2: Yes

6. Review Comments to the Author

You may also provide optional suggestions and comments to authors that they might find helpful in planning their study.

Reviewer #1: Thank you kindly to authors for making the suggested changes. I have no further comments at this time.

Reviewer #2: Thank you for clarifying your rationale for searching only MEDLINE and EMBASE and not other databases. You mention in your reviewer response that there was testing of the search strategy to establish that International Pharmaceutical Abstracts database did not need to be included in the search strategy for the review. Any testing of the strategy should be reported and discussed in the protocol. If EMBASE was deemed adequate for its coverage of the pharmaceutical literature, this should be explained and justified in the description of the search methods.

Thank you for providing more detail about the grey literature search strategy, including list of targeted websites. This will make the review methods more reproducible.

7. PLOS authors have the option to publish the peer review history of their article (what does this mean?). If published, this will include your full peer review and any attached files.

Reviewer #1: No

Reviewer #2: **Yes: **Gregory Laynor

---

## [Author Response · Author response to Decision Letter 1]

24 Jul 2024

July 25th, 2024

Dr. Udoka Okpalauwaekwe

Academic Editor

PLOS One

Re: PLOS ONE - Decision on Manuscript PONE-D-23-29159

Dear Dr. Okpalauwaekwe, 

We grateful for the further suggestions provided. We have made the suggested changes to the manuscript based on the suggestions from the reviewer’s comments. A point-by-point response to the comments is provided below. 

Sincerely, 

Araniy Santhireswaran

Reviewer Comments to the Author

Reviewer #1: 

1. Thank you kindly to authors for making the suggested changes. I have no further comments at this time.

Thank you for your comments, we appreciate the thoughtful suggestions you’ve shared.

Reviewer #2: 

1. Thank you for clarifying your rationale for searching only MEDLINE and EMBASE and not other databases. You mention in your reviewer response that there was testing of the search strategy to establish that International Pharmaceutical Abstracts database did not need to be included in the search strategy for the review. Any testing of the strategy should be reported and discussed in the protocol. If EMBASE was deemed adequate for its coverage of the pharmaceutical literature, this should be explained and justified in the description of the search methods.

Thank you for pointing this out, we agree that this should be mentioned in the manuscript. We have made appropriate changes to explain this in the manuscript.

Change: We understand that searching 3 databases is the standard for a scoping review and including a broad interdisciplinary database will improve our search. However, given that our search strategy is very specific we decided to conduct an extensive grey literature search instead of including a third broad interdisciplinary database. We tested our search in MEDLINE, EMBASE, Web of Science and Scopus, and decided that both MEDLINE and EMBASE were sufficient for our review. As interdisciplinary databases Web of Science and Pharmaceutical Abstracts did not bring in many relevant articles given our niche topic. We acknowledge that including only 2 databases is a potential limitation of our scoping review. Additionally, a supplementary hand-search of reference lists of included articles and any relevant reviews identified in screening will be conducted to identify any additional relevant articles that may have been missed by the search strategy.

2. Thank you for providing more detail about the grey literature search strategy, including list of targeted websites. This will make the review methods more reproducible.

We agree that this change will help with increasing reproducibility, thank you for your suggestion to elaborate on the grey literature search.

---

## [Decision Letter · Decision Letter 2]

21 Aug 2024

PONE-D-23-29159R2Impact of supply chain disruptions and drug shortages on drug utilization: a scoping review protocolPLOS ONE

Dear Dr. Santhireswaran,

Thank you for submitting your manuscript to PLOS ONE. After careful consideration, we feel that it has merit but does not fully meet PLOS ONE’s publication criteria as it currently stands. Therefore, we invite you to submit a revised version of the manuscript that addresses the points raised during the review process.

We look forward to receiving your revised manuscript.

Kind regards,

Udoka Okpalauwaekwe, MD, MPH, PhD

Academic Editor

PLOS ONE

Journal Requirements:

Reviewers' comments:

Reviewer's Responses to Questions

**Comments to the Author**

1. Does the manuscript provide a valid rationale for the proposed study, with clearly identified and justified research questions?

Reviewer #2: Yes

2. Is the protocol technically sound and planned in a manner that will lead to a meaningful outcome and allow testing the stated hypotheses?

Reviewer #2: Yes

3. Is the methodology feasible and described in sufficient detail to allow the work to be replicable?

Reviewer #2: Yes

4. Have the authors described where all data underlying the findings will be made available when the study is complete?

Reviewer #2: Yes

5. Is the manuscript presented in an intelligible fashion and written in standard English?

Reviewer #2: Yes

6. Review Comments to the Author

You may also provide optional suggestions and comments to authors that they might find helpful in planning their study.

Reviewer #2: The revision explains the rationale for only searching two databases (MEDLINE and EMBASE) and mentions that the search strategy was tested on other databases in order to decide not to include them. The data from the testing of the search strategy should thus be included as a supplement with the manuscript. The additional detail on the grey literature search strategy now included in the manuscript does make the methods more reproducible. The search methods section includes an incomplete sentence: "As interdisciplinary databases Web of Science and Pharmaceutical Abstracts did not bring in many relevant articles given our niche topic." Revising this sentence in the search methods section, and including the testing data for the search strategy, will make the protocol acceptable for publication.

7. PLOS authors have the option to publish the peer review history of their article (what does this mean?). If published, this will include your full peer review and any attached files.

Reviewer #2: **Yes: **Gregory Laynor

---

## [Author Response · Author response to Decision Letter 2]

16 Sep 2024

September 16th, 2024

Dr. Udoka Okpalauwaekwe

Academic Editor

PLOS One

Re: PLOS ONE - Decision on Manuscript PONE-D-23-29159

Dear Dr. Okpalauwaekwe, 

We grateful for the further suggestions provided. We have made the suggested changes to the manuscript based on the suggestions from the reviewer’s comments. A point-by-point response to the comments is provided below. In the manuscript with tracked changes we have highlighted all changes made since the original submission.

Sincerely, 

Araniy Santhireswaran

Reviewers' comments:

Reviewer's Responses to Questions

Reviewer #2: 

1. The revision explains the rationale for only searching two databases (MEDLINE and EMBASE) and mentions that the search strategy was tested on other databases in order to decide not to include them. The data from the testing of the search strategy should thus be included as a supplement with the manuscript.

Thank you for this suggestion. We acknowledge that only searching two databases is not the standard for a scoping review, and we did explore other databases including Web of Science and Scopus before reaching this decision. However, this decision process was not documented as it was a part of our preliminary brainstorming and was not done in a systematic manner. We were not aware that we should document this process since the PRISMA guidelines do not ask for this level of reporting. We understand how the wording in the manuscript can make it seem like it was a systematic process, so we have modified the wording for it to be clearer.

Change: We understand that searching 3 databases is the standard for a scoping review and including a broad interdisciplinary database will improve our search. However, given that our search strategy is very specific we decided to conduct an extensive grey literature search instead of including a third broad interdisciplinary database. We conducted preliminary explorations of our search in MEDLINE, EMBASE, Web of Science, Scopus and Pharmaceutical abstracts and decided that both MEDLINE and EMBASE were sufficient for our review.

2. The additional detail on the grey literature search strategy now included in the manuscript does make the methods more reproducible. The search methods section includes an incomplete sentence: "As interdisciplinary databases Web of Science and Pharmaceutical Abstracts did not bring in many relevant articles given our niche topic." Revising this sentence in the search methods section, and including the testing data for the search strategy, will make the protocol acceptable for publication.

Thank you again for the feedback on adding more detail on the grey literature search, we believe this has increased reproducibility of the review. Also, thank you for pointing out the incomplete sentence, we have modified it now.

Change: Given our niche topic, interdisciplinary databases such as Web of Science, Scopus and Pharmaceutical Abstracts did not bring in many relevant articles. We acknowledge that including only 2 databases is a potential limitation of our scoping review.

---

## [Decision Letter · Decision Letter 3]

9 Oct 2024

PONE-D-23-29159R3Impact of supply chain disruptions and drug shortages on drug utilization: a scoping review protocolPLOS ONE

Dear Dr. Santhireswaran,

Thank you for submitting your manuscript to PLOS ONE. After careful consideration, we feel that it has merit but does not fully meet PLOS ONE’s publication criteria as it currently stands. Therefore, we invite you to submit a revised version of the manuscript that addresses the points raised during the review process.

You are almost there. Just pay close attention to the comments from Reviewer 1 and you should be good to go.

We look forward to receiving your revised manuscript.

Kind regards,

Udoka Okpalauwaekwe, MD, MPH, PhD

Academic Editor

PLOS ONE

Journal Requirements:

Reviewers' comments:

Reviewer's Responses to Questions

**Comments to the Author**

1. Does the manuscript provide a valid rationale for the proposed study, with clearly identified and justified research questions?

Reviewer #2: Yes

2. Is the protocol technically sound and planned in a manner that will lead to a meaningful outcome and allow testing the stated hypotheses?

Reviewer #2: Yes

3. Is the methodology feasible and described in sufficient detail to allow the work to be replicable?

Reviewer #2: Yes

4. Have the authors described where all data underlying the findings will be made available when the study is complete?

Reviewer #2: No

5. Is the manuscript presented in an intelligible fashion and written in standard English?

Reviewer #2: Yes

6. Review Comments to the Author

You may also provide optional suggestions and comments to authors that they might find helpful in planning their study.

Reviewer #2: Two minor revisions/clarifications needed before the protocol is ready for publication:

Search methods: The search methods section gives a rationale for including two databases (MEDLINE and EMBASE) and a grey literature search instead of a broad interdisciplinary database. However, stating that "a broad interdisciplinary database will improve our search" is confusing. If it would improve the search to include a broad interdisciplinary database, why not do so? Instead, it could be clearer to just give the rationale for why the two disciplinary databases and targeted grey literature strategy were selected, rather than stating that the search would be improved by including an interdisciplinary database and then not doing so.

Data availability: The data availability statement states that all data from the review will be made available. However, it does not specify how or where the data will be made available.

7. PLOS authors have the option to publish the peer review history of their article (what does this mean?). If published, this will include your full peer review and any attached files.

Reviewer #2: **Yes: **Gregory Laynor

---

## [Author Response · Author response to Decision Letter 3]

10 Oct 2024

October 12th, 2024

Dr. Udoka Okpalauwaekwe

Academic Editor

PLOS One

Re: PLOS ONE - Decision on Manuscript PONE-D-23-29159

Dear Dr. Okpalauwaekwe, 

We grateful for the further suggestions provided. We have made the suggested changes to the manuscript based on the suggestions from the reviewer’s comments. A point-by-point response to the comments is provided below. In the manuscript with tracked changes we have highlighted all changes made since the original submission.

Sincerely, 

Araniy Santhireswaran

Reviewers' comments:

Reviewer's Responses to Questions

Reviewer #2: 

Two minor revisions/clarifications needed before the protocol is ready for publication:

1. Search methods: The search methods section gives a rationale for including two databases (MEDLINE and EMBASE) and a grey literature search instead of a broad interdisciplinary database. However, stating that "a broad interdisciplinary database will improve our search" is confusing. If it would improve the search to include a broad interdisciplinary database, why not do so? Instead, it could be clearer to just give the rationale for why the two disciplinary databases and targeted grey literature strategy were selected, rather than stating that the search would be improved by including an interdisciplinary database and then not doing so.

Thank you for this suggestion. We agree that this statement can be confusing. We have made appropriate changes. 

Changes: We understand that searching 3 databases is the standard for a scoping review. However, given that our search strategy is very specific we decided to conduct an extensive grey literature search instead of including a third broad interdisciplinary database.

2. Data availability: The data availability statement states that all data from the review will be made available. However, it does not specify how or where the data will be made available. 

Thank you for pointing this out. We have specified how the data will be made available in the data availability statement.

Changes: All data will be made available in the supplemental tables of the final published manuscript.

---

## [Decision Letter · Decision Letter 4]

22 Oct 2024

Impact of supply chain disruptions and drug shortages on drug utilization: a scoping review protocol

PONE-D-23-29159R4

Dear Dr. Santhireswaran,

We’re pleased to inform you that your manuscript has been judged scientifically suitable for publication and will be formally accepted for publication once it meets all outstanding technical requirements.

Kind regards,

Udoka Okpalauwaekwe, MD, MPH, PhD

Academic Editor

PLOS ONE

Additional Editor Comments (optional):

Reviewers' comments:

Reviewer's Responses to Questions

**Comments to the Author**

1. Does the manuscript provide a valid rationale for the proposed study, with clearly identified and justified research questions?

Reviewer #2: Yes

2. Is the protocol technically sound and planned in a manner that will lead to a meaningful outcome and allow testing the stated hypotheses?

Reviewer #2: Yes

3. Is the methodology feasible and described in sufficient detail to allow the work to be replicable?

Reviewer #2: Yes

4. Have the authors described where all data underlying the findings will be made available when the study is complete?

Reviewer #2: Yes

5. Is the manuscript presented in an intelligible fashion and written in standard English?

Reviewer #2: Yes

6. Review Comments to the Author

You may also provide optional suggestions and comments to authors that they might find helpful in planning their study.

Reviewer #2: Thank you for clarifying the rationale for the search methods and the specifics of the data availability.

7. PLOS authors have the option to publish the peer review history of their article (what does this mean?). If published, this will include your full peer review and any attached files.

Reviewer #2: **Yes: **Gregory Laynor

---

## [Editor Report · Acceptance letter]

24 Oct 2024

PONE-D-23-29159R4 

PLOS ONE

Dear Dr. Santhireswaran, 

I'm pleased to inform you that your manuscript has been deemed suitable for publication in PLOS ONE. Congratulations! Your manuscript is now being handed over to our production team.

Kind regards, 

on behalf of

Dr. Udoka Okpalauwaekwe 

Academic Editor

PLOS ONE